# Access to essential medicines for noncommunicable diseases during conflicts: The case cardiovascular diseases, diabetes and epilepsy in Northern Syria

Saleh Aljadeeah[1]*, Belen Tarrafeta[1], Samah Fahed[2], Karina Kielmann[3], Raffaella Ravinetto[1,4]

**1** Institute of Tropical Medicine, Antwerp, Belgium, **2** Independent Researcher, Brussels, Belgium, **3** Institute for Global Health & Development, Queen Margaret University, Musselburgh, United Kingdom, **4** School of Public Health, University of the Western Cape, Cape Town, South Africa

* saljadeeah@itg.be

## Abstract

Access to essential medicines is a critical component of healthcare. Conflicts severely disrupt pharmaceutical supply chains. This study examines the availability and prices of essential noncommunicable diseases (NCDs) medicines in Northern Syria, and explores the underlying factors contributing to medicine shortages and price variability. We applied a mixed-methods approach, combining a cross-sectional quantitative survey based on the World Health Organization/Health Action International (WHO/HAI) methodology with qualitative interviews. Medicine availability and price data were collected from public healthcare facilities and private pharmacies across Northeast and Northwest Syria. Availability was calculated as the percentage of facilities with the medicine in stock and categorized into four levels. Prices were reported using median price ratios. Qualitative data were collected from retailer pharmacists, representatives from non-governmental organizations (NGOs), and personnel working in medicine warehouses and wholesale distribution. Thematic content analysis was employed to analyze the qualitative data. The findings indicate low medicine availability, falling well below WHO targets. The mean availability was 45.5%. Of 28 medicines surveyed, 11 had somewhat high availability (50%–80%), 12 had low availability (30%–49%), and 5 had very low availability (<30%). Epilepsy medicines had the lowest availability, highlighting a particularly neglected area of care. Qualitative report suggested that the observed price variations were largely driven by the geopolitical tension, supply chain disruptions, and the absence of effective regulatory mechanisms. The conflict has caused or exacerbated shortages through the destruction of infrastructure, trade restrictions, and economic instability. Additionally, concerns over the quality of medicines were frequently reported. This study highlights challenges in accessing essential medicines in Northern Syria but also reveals that

which permits unrestricted use, distribution, and reproduction in any medium, provided the original author and source are credited.

**Data availability statement:** The data supporting this study cannot be made publicly available due to ethical considerations and restrictions imposed by the IRB. Sharing the complete dataset could compromise participant confidentiality and violate the commitments made during the informed consent process. Participants were clearly informed that their data would be securely stored, accessible only to the research team, and that personal information would remain confidential. To further protect participants' privacy, only non-identifiable codes were used when presenting quotations or related information in the manuscript. Public disclosure of the full dataset would conflict with these confidentiality assurances and pose potential risks to participants' privacy. Researchers interested in accessing de-identified data for academic purposes may direct their requests to the Institutional Review Board at the Institute of Tropical Medicine, Antwerp: IRB@itg.be. Data access requests will be reviewed in accordance with ethical and institutional guidelines.

**Funding:** This research was funded through a research grant of The King Baudouin Foundation (Fund Maurange, grant number: 2023-J1811240-229533 to SA). The funders had no role in study design, data collection and analysis, the decision to publish, or the preparation of the manuscript.

**Competing interests:** The authors have declared that no competing interests exist.

medicine supply systems, although disrupted, continue to operate through alternative and informal channels. Moving forward, efforts must prioritize coherent, context-sensitive policies that build on existing structures and human resources to rebuild a well-regulated and equitable system.

## Introduction

Essential medicines are a crucial intervention for the prevention, treatment, and management of diseases across all stages of care, until rehabilitation or palliative care [1]. The WHO defines essential medicines as those that satisfy the priority healthcare needs of the population, selected based on disease prevalence, evidence of safety and efficacy, and cost-effectiveness [1]. Access to essential medicines is a critical component of the right to health, intrinsically linked to principles of equality, non-discrimination, and transparency, as recognized in international human rights law [1,2]. It is integral to achieving Universal Health Coverage (UHC) and the Sustainable Development Goal (SDG) 3, which emphasizes healthy lives and well-being for all by 2030 [1,3]. The WHO has long been a global leader in promoting equitable access to essential medicines through initiatives such as the Essential Medicines List (EML), designed to prioritize medicines that address the health needs of the population [4]. Despite efforts, it was estimated in 2017 that nearly two billion people, primarily in low- and middle-income countries (LMICs), still lacked regular access to essential medicines, underscoring a persistent global challenge [5,6].

NCDs are among the leading causes of morbidity and mortality worldwide, responsible for 74% of all deaths annually [7]. These diseases, which include cardiovascular conditions, cancers, diabetes, epilepsy and chronic respiratory diseases, often result from a combination of genetic, environmental, and behavioral factors [7]. NCDs disproportionately affect LMICs, accounting for 77% of global NCD deaths, with 86% of premature deaths (before age 70) occurring in these settings [7]. Socioeconomic factors significantly contribute to this disparity, as poverty and limited access to healthcare exacerbate the burden of NCDs. Cardiovascular diseases remain a leading cause of NCD deaths, claiming 17.9 million lives annually [7]. Diabetes affects millions and is a significant contributor to kidney disease, while access to insulin and other treatments remain limited in many LMICs [7]. Epilepsy affects 50 million people globally, with a prevalence three times higher in LMICs compared to high-income countries [8]. Despite the efficacy of medicines in preventing and managing NCDs, millions struggle to access essential medicines due to a combination of factors including but not limited to high costs to households and individuals [8,9]. Addressing these challenges is critical to achieving global health goals, including SDG 3, which aims to reduce premature NCDs mortality by one-third by 2030 [7].

Syria, a country in the Eastern Mediterranean region has endured over a decade of conflict that has profoundly affected its health system. Before 2011, it was classified as a middle-income country and boasted some of the region's best health indicators [10]. Local pharmaceutical manufactory covered over 90% of local medicine

needs [10]. The conflict that began in 2011 marked a dramatic shift, resulting in widespread poverty, displacement, and the near-total collapse of the health and pharmaceutical system. By 2021, an estimated 13.4 million people required humanitarian assistance, including 6.7 million internally displaced persons (IDPs) [10], and about 70% of health workers fled the country [11]. The conflict also fragmented Syria into territories controlled by different authorities. The Syrian Ministry of Health oversaw the health system in territories held by the Assad government (about two-thirds of the country) [12], while the Autonomous Administration of North and East Syria managed health services in the northeast, facing limited international recognition and scarce resources [12,13]. In the Northwest, Turkish-controlled areas depended heavily on humanitarian organizations [12] (Fig 1). This fragmentation, together with repeated attacks on health facilities, left only half of facilities operational [10]. Although the end of the Assad regime in 2024 raised some hope for rebuilding the country, Syria remains deeply divided and instable [15]. Within this fragile context, NCDs such as cardiovascular diseases, diabetes, and epilepsy require particular attention, including continuous access to medicines: needs that humanitarian responses have often overlooked [10]. NCDs accounted for 45% of all deaths in Syria by 2021, reflecting a 40% increase since 2011 [16,17]. By 2016, about 60% of patients with diabetes requiring insulin therapy in Syria lacked access to the insulin [18,19]. Economic sanctions further disrupted Syria's pharmaceutical industry, resulting in shortages of essential medicines and a 50% increase in medicine prices [10].

The situation observed in Syria is not unique. Conflicts have profound indirect consequences on health, extending beyond harm caused directly by violence. They include the breakdown of health systems, which limits access to essential health services including medicines [20]. Availability and affordability, critical determinants of access to essential medicines, are growing challenges for healthcare systems globally, with medicines representing the largest household expense after food in many LMICs [4], but in conflict-affected areas, the situation is even more dire. The collapse of health systems and limited access to essential medicines turns otherwise manageable conditions like hypertension, diabetes, and epilepsy into life-threatening diseases [20]. Documenting the availability and prices of essential medicines in conflict-affected areas is vital not only to inform humanitarian interventions and rebuilding efforts, but also to provide historical accountability for the true cost of war [21,22]. Addressing these challenges is critical to ensuring that relief from serious health-related suffering becomes a priority in global efforts to mitigate the impacts of conflict [22]. To this purpose, this study aims to improve the knowledge base on access to medicines in conflict settings by measuring the availability and prices of a basket of essential medicines for hypertension, diabetes, and epilepsy in Northern Syria. It also explores stakeholder perspectives on availability, pricing mechanisms, and supply sources of these medicines.

## Materials and methods

This study adopted a cross-sectional and mixed-methods explanatory design, integrating both quantitative and qualitative data collection and analyses [23].

### Quantitative phase

Survey data were collected using the WHO/Health standardized methodology for measuring the availability and prices of essential medicines [24]. Availability and prices were recorded per package, and the package was used as the unit of analysis for both availability and price assessments. Given the unique constraints of Northern Syria's conflict context, the methodology was adapted to ensure feasibility and relevance, as described below.

**Sampling of medicine outlets.** Data on the availability and prices of essential medicines for the three selected NCDs were collected using adapted version from the WHO/HAI methodology [24]. This adapted methodology took into consideration the context of conflict in North Syria. First, the WHO/HAI methodology recommends collecting data from six geographical areas: a country's main urban center and five other administrative areas. However, given the operational constraints in Northern Syria, this was adjusted to focus on two main administrative areas, each further divided into two

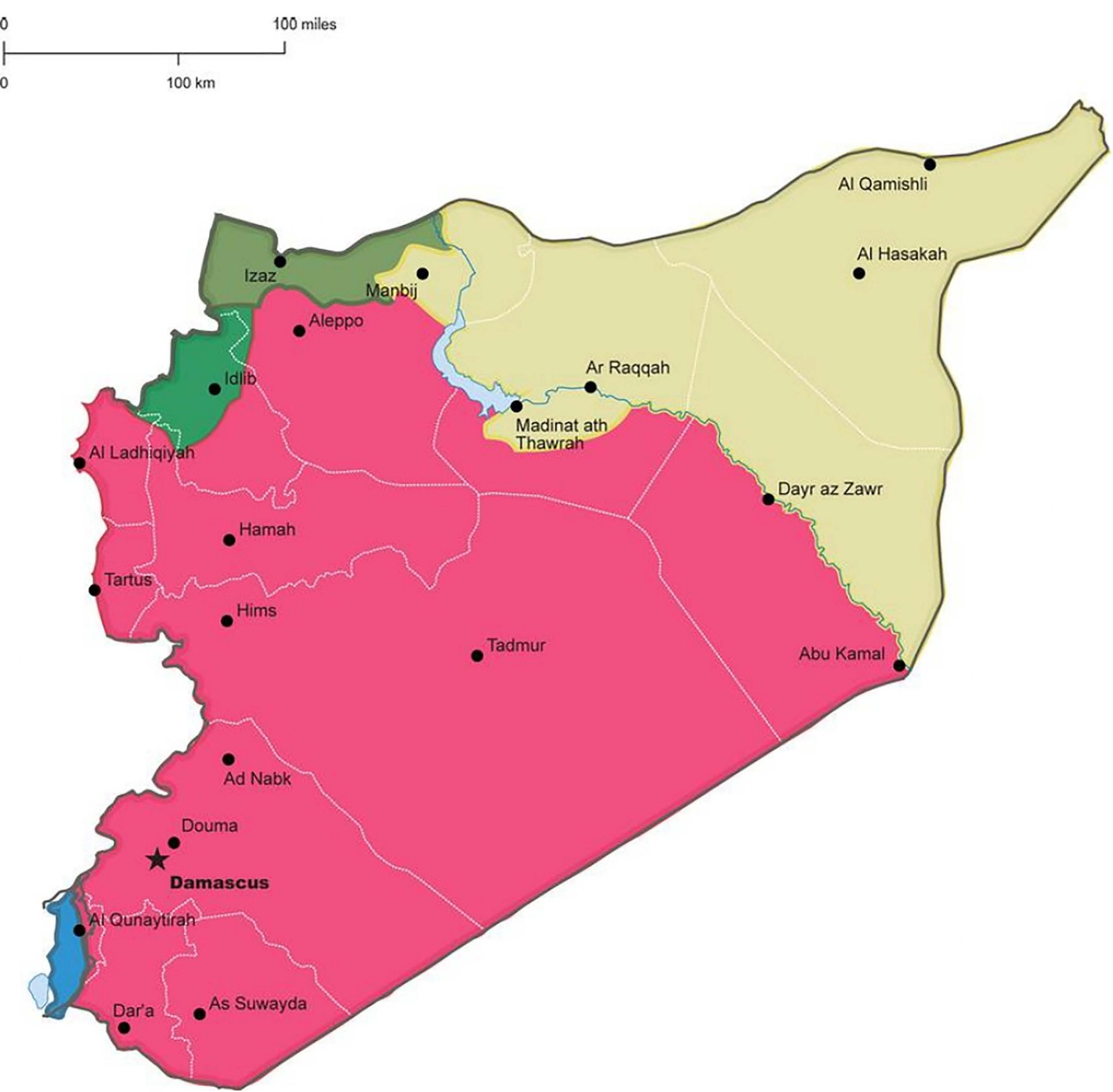

**Fig 1. Map of Syria, presenting Territorial control in Syria as of 2024.** Areas in pink represent regions controlled the Assad government, yellow areas are under the control of the Autonomous Administration of North and East Syria, green areas are controlled by opposition factions and Turkish-backed groups. Source: Noor Albeik, 2022. Additionally, an open access regularly-updated map can be found at https://syria.liveuamap.com/ [14].

subregions. Second, the WHO/HAI guidelines advise examining medicine outlets in both the public and private sectors, including primary healthcare centers and government hospitals in the public sector and licensed pharmacies and drug stores in the private sector [24]. However, in Northern Syria, the formal classification of healthcare facilities was not feasible due to the absence of many typical public sector institutions, such as government-run hospitals and primary care centers; these gaps were partially filled by humanitarian international NGOs. Third, the definition of "licensed" pharmacies and drug stores varied across the study area. Local authorities in Northern Syria often did not license or register retail pharmacies and lacked the capacity to regulate their practices. Therefore, we considered any retail pharmacy with a practicing pharmacist as eligible for the study. In each of the survey subregions (two administrative areas, each with two subregions), the sampling strategy overall included: 1) One hospital, and two-to-four primary healthcare centers, supervised by either local governance or INGOs, representing the public sector. 2) Five private pharmacies, representing the private sector. This adjusted sampling framework ensured that the availability and price survey covered a sample that is likely representative of healthcare facilities in this particular context.

**Selection of surveyed essential medicines.** The selection of medicines for the survey was guided by the WHO 23rd Model List of Essential Medicines for adults [25], and the tenth edition of the Syrian National Essential Medicines List (2019) [26]. The selection of medicines for cardiovascular diseases and diabetes was based on three complementary strategies: prior evidence of common prescription patterns [11], inclusion in the tenth Syrian National Essential Medicines List, and input from the advisory board (as described below) and HAI expert. For epilepsy essential medicines, the selection was guided by the tenth Syrian National Essential Medicines List, by the advisory board, by the HAI expert inputs, as no data on medicine use or prescribing patterns for epilepsy in Syria were available. In contrast to the WHO/HAI methodology, the survey focused exclusively on generic medicines, given the limited availability of innovators (sometimes called 'branded' medicines) in the study area [27].

**Recruitment and data collection.** A core sample of participants was purposively recruited through the principal investigator's professional network in Northern Syria. Additionally, snowball sampling was used to identify and recruit further participants. The data collection process was adapted from the WHO/HAI standardized methodology [24], to be applicable to the unique challenges of this context. Unlike the WHO/HAI recommendation to collect data in person, data for this study were collected remotely, by reaching out to representatives of health facilities online, and sharing with them online the data collection forms. This approach ensured feasibility while maintaining accurate data collection process. All facilities whose representatives consented to participate were sent the standardized data collection forms by email. The respondents were guided through the process during virtual sessions with the principal investigator, ensuring consistency and accuracy in their responses. Data were collected over the period from 02. January to 15. March 2024. A medicine was considered "available" if the respondent confirmed that it was in stock at the time of their participation. Price data, defined as the selling price of medicines to end-users, were collected only for those medicines reported to be available at the time of the survey. Information about medicines prices was not collected from facilities where medicines could be dispensed for free, while it was collected from the private sector.

**Data analysis.** Availability: Availability was defined according to the WHO/HAI methodology as the percentage of surveyed facilities with a given medicine in stock on the day of the survey. Availability levels were categorized as follows: Very low: < 30%, Low: 30%–49%, Somewhat high: 50%–80%, High: > 80% [24].

Prices: Median price ratios (MPRs) is a measure suggested in the WHO/HAI guidelines used to facilitate international comparisons of medicine prices. It is calculated by expressing the median local price of a package of medicine as a ratio relative to an international reference price. An MPR greater than 1 indicates that the local price exceeds the reference price, while an MPR less than 1 signifies that the local price is lower [24]. MPR were not calculated in this study, because the international price reference guidelines have not been updated since 2015 [28], making it obsolete for comparisons. Instead, the study adhered to HAI's more recent recommendation to report prices of lowest-priced generics using median values, expressed in USD (conversion rate: 1 USD = 15,000 Syrian Pounds).

## Qualitative interviews

**Participant selection and recruitment.** Qualitative interviews were conducted by the principal investigator of this study (SA) with individuals directly or indirectly involved in providing access and supplying of essential medicines in Northern Syria. Participants included retailer pharmacists, representatives from non-governmental organizations (NGOs), and personnel working in medicine warehouses and wholesale distribution. Initially, the principal investigator leveraged his professional network to identify potential participants. This was complemented by snowball sampling, where initial participants referred other eligible individuals within their networks. To ensure a diverse sample, a purposive sampling strategy was implemented. This strategy aimed to include participants with varying demographic characteristics, including age, sex, and geographical location (Northeast and Northwest of Syria), so as to capture a broad spectrum of experiences and perspectives. Prior to each interview, all participants provided verbal consent, which was documented through audio recording.

**Data collection.** Data collection for the qualitative phase involved semi-structured interviews (S1 Appendix). The interviews were designed to explore the needs, perceptions, and experiences of different stakeholders concerning medicine access and supply within the specific context of Northern Syria. Key areas of inquiry included the supply and sources of NCDs essential medicines, their availability, and the mechanisms influencing their pricing. Interviews were conducted remotely via voice calls with secured online platforms, in Arabic. A topic guide was developed from the study's research questions and discussed with the research advisory board provided structure for each interview. Interviews were conducted over the period from 15 May 2023 to 30 March 2024 All interviews were audio-recorded, with respondents' consent, and transcribed verbatim in Arabic by SA and SF, both of them are native Arabic speakers. The full transcripts were not translated into English.. SA and SF conducted the thematic analysis of the Arabic transcripts including development of a set of pertinent codes. The identified codes were then translated into English to enable all co-authors, including those not involved in data collection, to actively engage in the analytic process. Co-authors reviewed and provided feedback on the defined codes, categories, and emerging themes. Final themes were refined through iterative feedback until consensus was reached. The coding and thematic analysis were conducted using Microsoft Excel.

The process of recruitment and interviewing continued until data saturation was reached. While the definition and application of saturation can vary [29], this study adopted the definition of saturation as the point where no new substantive themes or information relevant to the research questions were observed in the data [30]. Specifically, saturation was assessed at the thematic level, ensuring that the identified themes comprehensively addressed the research questions and that further interviews were unlikely to yield significant new insights.

**Data analysis.** Thematic content analysis was employed to analyze the transcribed interviews data. An inductive approach was chosen, allowing codes, categories, and themes to be identified directly from the data. This data-driven method facilitated the identification of patterns and insights without imposing pre-conceived theoretical frameworks or hypotheses [31]. The inductive approach was deemed particularly suitable for this exploratory research, enabling the study to remain firmly grounded in the participants' actual experiences and perspectives [32]. This approach enhanced the credibility and accuracy of the findings, providing a nuanced understanding of the challenges related to accessing essential medicines within the specific research context.

**Research advisory board.** At the protocol-writing stage, we established a research advisory board comprising members from the concerned communities and the Syrian diaspora in Europe. The board included 15 members, representing a diverse group of healthcare providers, patients, and health activists. Care was taken to ensure diversity in age, gender, and educational backgrounds, to create an inclusive and balanced platform for discussion and inputs. The research advisory board provided a collaborative and open space for critical feedback and worked entirely online to accommodate the geographical dispersion of its members. The board was actively involved at various stages of the project. Their consultation began during the early stages of defining the study's focus. They contributed to subsequent steps, including the development of study materials, finalization of protocols, informed consent strategies, participant

recruitment approaches, data collection strategies, and data analysis. The advisory board's contributions were critical to ensuring that the study was contextually relevant, ethically sound, and aligned with the needs and perspectives of the communities it aimed to serve.

**Ethical considerations.** The study protocol was submitted for review and approved by the Institutional Review Board (IRB) of the Institute of Tropical Medicine (Reference: 1660/23, 16.03.2023), which has expertise in reviewing research conducted in fragile settings. To compensate the absence of a local research ethics committee, the Research Advisory Board described above provided critical input on different aspects with ethics implications, such as the feasibility and acceptability of remote digital data collection, participant and data collector security, emergency response strategies, and risks of re-traumatization and stigmatization. These insights were integrated into the protocol to ensure responsiveness to local needs. Additionally, as part of their review of the protocol, the IRB specifically asked the research team to identify psychological support services (e.g., NGO-operated programs) to which participants who expressed any form of distress could be referred, recognizing the limitations of such services in Syria. This measure was implemented, and where formal support was unavailable due to the local circumstances, the use of culturally appropriate self-help materials, such as those developed by the WHO [33], was applied as an alternative.

## Results

### Quantitative results

A total of 37 healthcare facilities participated in the availability and prices survey. These included 20 private retail pharmacies, representing the private sector, and 17 healthcare facilities (4 hospitals and 13 primary care centers) supervised by either local governance or INGOs, representing the public sector.

**Availability of essential medicines.** The mean availability of the surveyed NCDs medicines was 45.5% (low availability). Eleven out of the 28 medicines included in the survey had somewhat high availability (50%-80%), 12 medicines had low availability (30%-49%) and 5 medicines had very low availability (<30%). None of the surveyed medicines had a high availability (80% or more). The mean availability of the surveyed NCDs essential medicines was lower in Northeast Syria (42.3, low availability) compared to the Northwest (49.2, low availability). Aspirin was the most available medicine among those surveyed (71.3, somewhat high) and Oxcarbazepine (antiepileptic) was the least available one (10.5, very low).

Out of the 19 medicines for CVDs, four had very low availability (less than 30%), six had low availability (30–49%), and nine had somewhat high availability (50–80%) (Table 1).

Three out of the five medicines for diabetes had low availability (30–49%), and two had somewhat high availability (50–80%). One out of the four antiepileptic medicines had very low availability (less than 30%) and the other three had low availability (30–49%). (Table 1).

In the public sector, the mean availability of CVDs essential medicines was 39.3% (low availability); for diabetes medicines the mean availability was 34.1% (low availability) and for epilepsy medicines the mean availability was 19.1% (very low availability). In the private sector, the mean availability of CVDs medicines was 54.7% (somewhat high), while for diabetes medicines it was 57.0% (somewhat high) and for epilepsy medicines it was 43.7% (low). Mean availability of CVDs medicines in the Northeast was 46.0% (low), that is lower than in the Northwest (49.8, also low). Similarly, the availability of diabetes medicines was lower in the Northeast (41% compared to 54.1 in the Northwest), and it was the same for epilepsy (26.2 in the Northeast vs 39.7 in the in the Northwest). (Fig 2).

**Pricing of essential medicines.** Twenty-four out twenty-eight medicines were found in ≥ 4 private pharmacies and hence eligible, based on our methodology, for measurement of the median price (Table 2).

The median prices varied between Northeast and Northwest Syria. Medicines median prices were generally higher in Northwest Syria compared to Northeast Syria. For example, a dispensing unit of Captopril 25 mg (Package: 20 tablets)

**Table 1. Availability of NCDs essential medicines.**

| Essential medicines by therapeutic class | Northeast (%) | Northwest (%) | Private sector (%) | Public sector (%) | Total (%) |
|---|---|---|---|---|---|
| **1.Cardiovascular diseases** | | | | | |
| **1.1 Antihypertensives** | | | | | |
| Captopril 25 mg 20 tab | 65.0 | 64.7 | 70.0 | 58.8 | 64.6 |
| Enalapril 10 mg/20 tab | 55.0 | 64.7 | 60.0 | 58.8 | 59.6 |
| Lisinopril 10 mg/28 tab | 15.0 | 17.6 | 20.0 | 11.8 | 16.1 |
| Amlodipine 5 mg/30 tab | 70.0 | 70.6 | 65.0 | 76.5 | 70.5 |
| Amlodipine + Telmisarta 40 + 5/30 tab | 30.0 | 11.8 | 20.0 | 23.5 | 21.3 |
| Bisoprolol 5 mg/20 tab | 60.0 | 64.7 | 65.0 | 58.8 | 62.1 |
| Metoprolol 50 mg/40 tab | 45.0 | 64.7 | 65.0 | 41.2 | 54.0 |
| Nebivolol 5 mg/30 tab | 20.0 | 35.3 | 40.0 | 11.8 | 26.8 |
| Losartan + Hydrochlorothiazide 50 + 12.5 mg/30 tab | 30.0 | 52.9 | 60.0 | 17.6 | 40.1 |
| Valsartan + Hydrochlorothiazide 80 + 12.5 mg/30 tab | 40.0 | 52.9 | 65.0 | 23.5 | 45.3 |
| Diuretics | | | | | |
| Furosemide 40 mg/20 tab | 70.0 | 70.6 | 70.0 | 70.6 | 70.3 |
| Hydrochlorothiazide 25 mg/30 tab | 50.0 | 41.2 | 45.0 | 47.1 | 45.9 |
| Spironolactone 25 mg/20 tab | 50.0 | 58.8 | 65.0 | 41.2 | 53.8 |
| **1.2 Lipid-Lowering Agents** | | | | | |
| Atorvastatin 20 mg/30 tab | 40.0 | 58.8 | 70.0 | 23.5 | 48.1 |
| Rosuvastatin 20 mg/30 tab | 35.0 | 47.1 | 65.0 | 11.8 | 39.7 |
| Simvastatin 20 mg/ 30 tab | 20.0 | 5.9 | 10.0 | 17.6 | 13.4 |
| **1.3 Antiplatelet agents** | | | | | |
| Aspirin 81 mg/50 tab | 75.0 | 70.6 | 75.0 | 64.7 | 71.3 |
| **1.4 Nitrates** | | | | | |
| Isosorbide dinitrate 5 mg/40 tab | 60.0 | 35.3 | 45.0 | 52.9 | 48.3 |
| **1.5 Antiarrhythmics** | | | | | |
| Amiodarone 200 mg/20 tab | 45.0 | 58.8 | 65.0 | 35.3 | 51.0 |
| **2. Diabetes** | | | | | |
| Glibenclamide 5 mg/30 tab | 30.0 | 58.8 | 45.0 | 41.2 | 43.7 |
| Gliclazide 80 mg/30 tab | 50.0 | 52.9 | 65.0 | 35.3 | 50.8 |
| Metformin 500 mg/40 tab | 60.0 | 70.6 | 70.0 | 58.8 | 64.9 |
| Metformin + Sitagliptin 1000 + 50 mg/30 tab | 35.0 | 41.2 | 60.0 | 5.9 | 35.5 |
| Insulin human mixtard 30 + 70 IU/ml/1 vial | 30.0 | 47.1 | 45.0 | 29.4 | 37.9 |
| **3. Epilepsy** | | | | | |
| Carbamazipine 200 mg/50 tab | 30.0 | 58.8 | 60.0 | 23.5 | 43.1 |
| Oxcarbazepine 300 mg/30 tab | 15.0 | 5.9 | 15.0 | 5.9 | 10.5 |
| Leveteiracetam 500 mg/30 tab | 30.0 | 41.2 | 50.0 | 17.6 | 34.7 |
| Valproic acid 500 mg/40 tab | 30.0 | 52.9 | 50.0 | 29.4 | 40.6 |

Availability was described using the range: Very low: < 30% (red); Low: 30–49% (orange); and Somewhat high: 50–80% (green).

was priced at 1.4 USD [1.07–1.70] in Northwest Syria, whereas its price in Northeast Syria ranged from 0.5 to 0.60 USD, which is significantly lower. Similarly, a dispensing unit of Atorvastatin 20 mg (Package: 30 tablets) showed a higher price range in Northwest Syria (1.4 USD [1.1–1.7]) compared to Northeast Syria (1.1–1.3 USD). The degree of price variation within each region was also notable. Within Northwest Syria, price variation was particularly evident for Aspirin 81 mg (Package: 50 tablets), which ranged from 0.8 to 1.9 USD (Table 2).

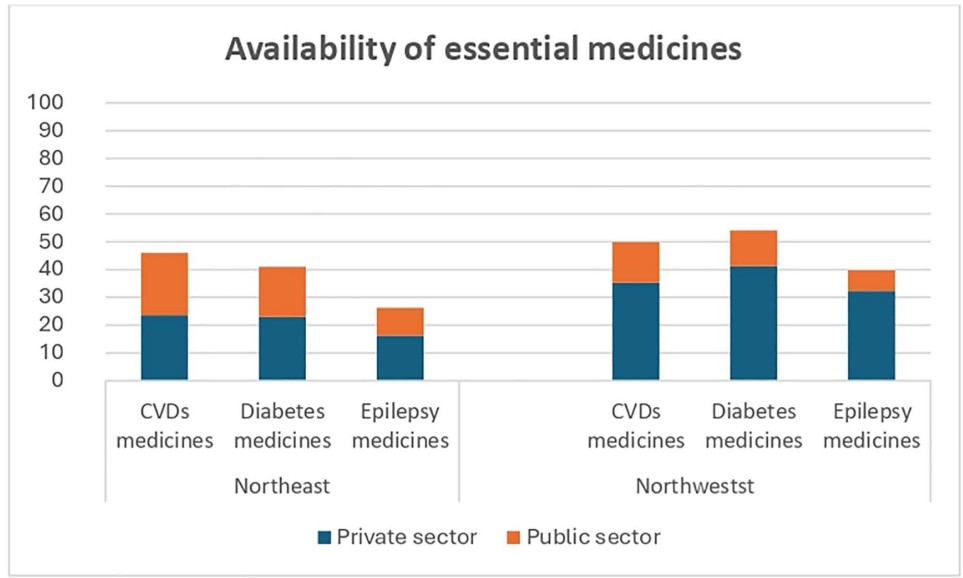

**Fig 2. Availability of NCDs essential medicines by region and sector.**

## Qualitative results

We conducted interviews with 15 participants, including retailer pharmacists, NGOs and pharmaceutical wholesale employees (Table 3). The interviews lasted between 17 and 99 minutes (median 34).

The qualitative findings highlight the complex pharmaceutical market in Northeast and Northwest Syria. They also provide context for the survey findings. We identified four themes in the interviews conducted including: sources and supply of medicines, their availability, pricing dynamics, and concerns about medicine quality.

**Reported sources and supply of medicines.** Medicines in Northeast and Northwest Syria reportedly came from a variety of sources, reflecting the region's fragmented healthcare system and the impact of conflict-related factors on medicine supply chains. Participants described four main sources of medicines: (1) medicines produced in the Assad government-controlled areas (2), medicines imported through neighboring countries such as Turkey and Iraq (3), so-called unregulated 'tourist medicines' that enter the region through informal suppliers and trade routes, and (4) locally produced medicines from small-scale often unregulated pharmaceutical factories operating in the region.

In Northeast Syria, one participant described: "*The majority of the medicines here are Syrian-made in government-controlled areas, and the rest were mainly imported from abroad.*" (Participant 1, Northeast Syria). Northwest Syria depended more on Turkish imports, with Turkish pharmaceutical companies supplying medicines that are often perceived to be of better quality, but are priced higher than alternatives from other sources. In both regions, private wholesalers played a central role in storing and selling medicines to smaller distributors and pharmacies.

The supply of medicine in both of the Northeast and the Northwest regions was deeply affected by political instability, security risks, and logistical barriers regardless of the supply source. As a participant stated: "*Medicines must pass through multiple checkpoints and customs processes when transported from government-controlled areas to Northeast Syria. This leads to delays and increased costs*" (Participant 2, Northeast Syria). This is particularly evident in Northwest Syria, where smuggling networks play a significant role in medicine distribution: "*The smuggling companies coordinate between government-affiliated military factions and factions in opposition-controlled areas, such as the Syrian National Army, to facilitate the transportation process.*" (Participant 3, Northwest Syria).

**Table 2. Median price [the 25th –75th Percentile] of lowest priced generic essential medicines packages in USD (private pharmacies, n = 20).**

| List of EMs available in at least four retail pharmacies | Northeast Syria | Northwest Syria |
|---|---|---|
| **Cardiovascular diseases** | | |
| **1.1 Antihypertensives** | | |
| Captopril 25 mg (20 tablets) | 0.5, 0.5, 0.6 | 1.4 [1.1–1.7] |
| Enalapril 10 mg (20 tablets) | | 1.0 [0.6–1.0] |
| Lisinopril 10 mg (28 tablets) | | 1.0, 1.9 |
| Amlodipine 5 mg (30 tablets) | 0.6, 0.7 | 0.9 [0.8–1.0] |
| Bisoprolol 5 mg (20 tablets) | 0.6, 0.6 | 0.8 [0.8–0.9] |
| Metoprolol 50 mg (40 tablets) | 1.1 | 1.3 [1.2–1.5] |
| Nebivolol 5 mg (30 tablets) | 1.6 | 1.8 [1.7–2.1] |
| Losartan + Hydrochlorothiazide 50 + 12.5 mg (30 tablets) | 0.9 | 1.1 [1.0–1.2] |
| Valsartan + Hydrochlorothiazide 80 + 12.5 mg (30 tablets) | 1.3, 1.3 | 1.5 [1.1–1.6] |
| Diuretics | | |
| Furosemide 40 mg (20 tablets) | 0.5 | 0.7 [0.6–0.8] |
| Hydrochlorothiazide 25 mg (30 tablets) | | 0.8 [0.7–1.2] |
| Spironolactone 25 mg (20 tablets) | 0.8 | 1.0[0.9–1.0] |
| **1.2 Lipid-lowering agents** | | |
| Atorvastatin 20 mg (30 tablets) | 1.1, 1.3, 1.3 | 1.4 [1.1–1.7] |
| Rosuvastatin 20 mg (30 tablets) | 1.6 | 1.4 [1.0–1.5] |
| Simvastatin 20 mg (30 tablets) | 1.7 | |
| **1.3 Antiplatelet agents** | | |
| Aspirin 81 mg (50 tablets) | 1.2, 1.8, 1.8 | 1.4 [0.8–1.9] |
| **1.4 Nitrates** | | |
| Amiodarone 200 mg (20 tablets) | 2.2 | 2.3 [2.1–2.5] |
| **1.5 Antiarrhythmics** | | |
| Isosorbide dinitrate 5 mg (40 tablets) | | 1.1 [0.8–1.2] |
| **Diabetes** | | |
| Glibenclamide 5 mg (30 tablets) | | 0.6 [0.5–1.0] |
| Metformin 500 mg (40 tablets) | 1.1 | 1.3 [1.1–1.4] |
| Gliclazide 80 mg (30 tablets) | 0.9 | 1.15 [1.0–1.3] |
| Insulin human mixtard 70 + 30 IU (1 vial) | | 3.3 [3.1–4.2] |
| Metformin + Sitagliptin 1000 + 50 mg (30 tablets) | 1.9 | 2.2 [2.0–2.8] |
| **Epilepsy** | | |
| Carbamazipine 200 mg (50 tablets) | 2.9 | 2.0 [2.0–3.5] |
| Valproic acid 500 mg (40 tablets) | | 3.0 [2.9–3.3] |
| Oxcarbazepine 300 mg (30 tablets) | 2.5 | 1.5 |
| Levetiracetam 500 mg (30 tablets) | 3.7 | 2.9 [2.4–3.1] |

Wholesalers are subject to arbitrary customs fees, and shipments can be delayed for months due to political tensions and security concerns. Furthermore, armed groups often demanded payments for safe passage outside official and unofficial checkpoints, adding additional financial burdens on suppliers and pharmacies. Medicines requiring cold chain such as insulin, are particularly affected in this complex scenario due to the specific transportation requirements necessary to maintain appropriate temperatures. These challenges are further aggravated by the unreliable electricity as a participant highlighted: "*Insulin requires refrigerators and stable electricity, which is challenging in this region*." (Participant 4, Northwest Syria).

**Table 3. Qualitative study participants.**

| Participants | N | | |
|---|---|---|---|
| | **Retailer Pharmacists** | **NGO employees** | **Wholesale employees** |
| **Median age** | 37 Years | 35.5 Years | 40.5 Years |
| **Sex** | | | |
| Male | 5 | 2 | 4 |
| Female | 2 | 2 | 0 |
| **Location** | | | |
| Northeast | 4 | 2 | 3 |
| Northwest | 3 | 2 | 1 |
| **Total** | 15 | | |

Participants also reported the widespread use of digital tools, particularly WhatsApp, to obtain up-to-date information about medicine sources, prices, and availability from suppliers, and to coordinate with other pharmacists. This use of mobile technologies facilitated real-time information exchange and informal coordination, which proved particularly valuable in a context marked by fragmented supply chains and rapidly changing availability and prices of medicines.

International and local humanitarian organizations played an essential role in supplying medicine in Northern Syria. However, the supply of medicine relied on available funding, which is normally tied to time-bound projects, making it inconsistent and often failing to meet the needs of patients with chronic illnesses: "*Several international organizations provide medicine, particularly for diabetes and hypertension. However, their support is limited and cannot meet the full demand in the region*" (Participant 2, Northeast Syria).

**Participants' observations on availability of NCDs essential medicines.** Participants provided multiple reasons to explain why the availability of essential medicines for NCDs is low in Northern Syria. These reasons included high import costs, supply chain disruptions caused by conflict and sanctions, and prioritization of acute care over chronic conditions by different stakeholders. Many humanitarian organizations prioritized infectious disease management over chronic conditions, exacerbating the issue: "*Many organizations are focusing on infectious diseases as part of public health strategies to protect communities from outbreaks. However, Syria's current situation demands a dual approach that addresses both the prevention of infectious diseases and the treatment of other illnesses.*" (Participant 5, Northwest Syria).

Among NCD medicines, epilepsy medicines (antiseizures) were of particularly low availability. Some participants noted that epilepsy medicines such as sodium valproate and carbamazepine were not produced in large quantities in the Assad government-controlled areas and were often neglected by suppliers, leading to common shortages and stockouts. The lack of regulation of medicines prices in the local market allowed some retail pharmacists to set high prices, particularly for medicines in very low availability, such as epilepsy medicines, which resulted in market monopoly further limiting access: "*Epilepsy medicines, like sodium valproate, are distributed in very limited quantities, approximately 15 or 20 packs, to the entire region. Sometimes they are available without notice, and two or three pharmacists purchase all of them, leaving the rest without any stock.*" (Participant 6, Northeast Syria). As a result, patients with epilepsy often had to search multiple pharmacies to find their medicines or relied on inconsistent supplies from humanitarian organizations.

**Participants' observation on prices of NCDs essential medicines.** Medicine prices varied significantly between different pharmacies due to the absence of pricing regulation and/or endorsement mechanisms. In Northeast Syria, retailer pharmacists were free to set the profit margin, while in Northwest Syria, authorities often issued lists of medicines with fixed prices in an attempt to regulate prices, but often failed to enforce them: "*There is significant*

*variation in prices. Each pharmacy sets its own profit margins, which can range from 25% to 40%.*" (Participant 7, Northwest Syria).

According to the respondents, transportation costs, customs fees, and smuggling expenses have further inflated medicine prices. Participants described how the cost of moving medicines from the Assad government-controlled areas into opposition-held regions significantly increased the final price paid by consumers: "*Transporting one cubic meter of medicines from government-controlled areas to our region costs about $900, significantly increasing the price of medicines in our market.*" (Participant 8, Northwest Syria).

Participants referred that the economic crisis and international sanctions have led to increased medicine prices, particularly as local pharmaceutical manufacturers struggled with the high cost of raw materials, which are imported in US dollars while medicines were sold in Syrian pounds: "*The rising dollar exchange rate has caused significant disruptions. Since pharmaceutical factories purchase raw materials in dollars but sell their products in Syrian pounds, they often halt production during periods of high exchange rate volatility.*" (Participant 9, Northeast Syria).

Participants explained how the geopolitical situation resulting from the conflict contributed to higher prices of essential medicines in Northwest Syria compared to the Northeast: "*In Northeast Syria, the $500 transportation cost is paid to the military factions controlling the checkpoints in government-controlled areas. In our region (*Northwest Syria*), the cost is higher because an additional $400 is paid to factions in opposition-controlled areas, such as the Syrian National Army, to secure the shipment through smuggling points*" (Participant 3, Northwest). The prices of NCDs medicines in this context were rather high compared to other medicines such as antibiotics. A participant explained this as following: "*These medicines* (for NCDs) *are highly demanded in the market, but their availability is limited. They are often more expensive compared to other medicines.*" (Participant 10, Northeast Syria).

**Perceived quality of NCDs essential medicines.** Although the study primarily focused on the availability and pricing of some essential NCD medicines, participants frequently raised concerns about their quality. Information about medicines quality were identified as a theme during the qualitative data analysis. Many participants raised concerns about the quality of medicines, particularly those that pass through illegal supply routes. Inadequate transportation conditions, particularly when linked to smuggling and poor storage conditions explained the poor quality of some medicines in the market: "*Warehouses suffer from high humidity levels, lack dehumidifiers, and do not measure temperature or humidity levels, except for a few reputable warehouses managed by pharmacists.*" (Participant 11, Northwest Syria).

The challenging market conditions appear to hinder suppliers from implementing consistent quality assurance management systems, especially as they operate without regulatory oversight and lack access to quality control laboratories: "*Medicines are not subject to any drug monitoring, as there are no laboratories to test their quality. As a result, low-quality or ineffective medicines are frequently sold in the market. For example, we recently discovered that a batch of aspirin originated from India was ineffective and unfit for use, despite being sold by a prominent warehouse in the region*" (Participant 3, Northwest).

Several participants highlighted the interplay between the quality, availability, and prices of medicines. A participant emphasized the challenge of balancing quality and affordability: "*The challenge is balancing quality and affordability. Patients demand cheaper medicines, but I prioritize sourcing high-quality medicines*" (Participant 12, Northwest). Another participant highlighted a decline in the quality of medicines produced in government-controlled areas: "*The quality of medicines produced in the government areas has also declined in recent years due to the economic crisis and the lack of oversight over manufacturing processes*" (Participant 13, Northeast Syria). Participant 11 highlighted the effect of market concentration by certain wholesalers as a factor driving high prices during certain periods of time, until other suppliers replenish their stocks.: "*Sometimes, we face price fluctuations due to monopolization by certain warehouses, which can drive up prices until the medicines are restocked in the market.*" (Participant 11, Northwest Syria). These responses demonstrate how issues of quality, availability, and pricing are closely linked, with participants recognizing the complex trade-offs involved in ensuring access to essential medicines.

## Discussion

To the best of our knowledge, this study provides the first estimates of the availability and prices of essential medicines in Northern Syria after the onset of the conflict, using the standard methodology of the HAI/WHO, and utilizing a mixed-methods approach that has involved both qualitative and quantitative data. By drawing on both qualitative interviews with the quantitative analysis of availability and pricing data, the study offers a comprehensive view of the challenges and barriers to accessing essential medicines, and of the supply pathways for medicines in Northern Syria. The study's key findings highlight the low availability of our basket of essential medicines for NCDs, considering both the supply through humanitarian aid as well as retail pharmacies. This indicates a clear link between the unmet needs, the geopolitical situation and other conflict-related factors. First, the study revealed that the availability of these medicines was well below the minimal targets set by the WHO, with most medicines being of low availability. Second, it reported a significant price variation between the Northeast and Northwest of Syria, and a significant price variation among suppliers within each of these regions, and fluctuations of prices over time.

The low availability of essential medicines is not surprising, considering the multifaceted challenges arising from the conflict. The devastation of infrastructure, destruction of health facilities, disruption of supply chains, and the scarcity of qualified healthcare professionals have severely undermined the pharmaceutical supply chains. These factors have led to widespread shortages and irregular supply of essential medicines, including those required for the management of NCDs. The findings are consistent with other studies conducted in conflict-affected countries, such as Yemen and Afghanistan, where supply of medicine was constrained by similar factors leading to low availability of essential medicines [34,35], and to heightened morbidity or mortality - which do not capture the attention of international stakeholders, perhaps being fatalistically perceived as one more side effect of the war. For instance, the average availability of furosemide was 70% in Syria, 70% in Yemen, 61% in Afghanistan, and 71% in Gaza, while the availability of atorvastatin was 48% in Syria, compared to 57% in Yemen, 39% in Afghanistan, and 28% in Gaza [34–36]. Beyond conflict-affected settings, evidence from low- and middle-income countries also reveals considerable variability in the availability of essential medicines for NCDs, with some medicines reported to have high availability while others were reported to have low or very low. For example, in Tanzania, the availability of metformin 500 mg was reported to be 100% while in our study it was 64.9%, yet gliclazide 80 mg had 0% availabity in the surveyed facilities in Tanzania compared with 50.8% availability in our study [4]. A study reported that the availability of carbamazepine 200 mg tablets in Ethiopia was 82.8%, whereas it was 43.1% in our study [8]. In Kenya, the availability of amlodipine 5 mg was 31.6%, bisoprolol 5 mg 0.7%, furosemide 40 mg 71.7%, and glibenclamide 5 mg 66.1%, compared to 70.5%, 62.1%, 70.3%, and 43.7%, respectively, in our study [37]. These data suggest that while certain medicines may achieve high availability in some LMICs contexts, others remain consistently of low availabity, reflecting systemic constraints such as weak supply chains and limited regulatory capacity. However, these comparisons should be interpreted with great caution due to several key differences. First, when looking at conflict contexts only, the data were collected from different types of facilities: our study surveyed public hospital pharmacies, primary healthcare centers run by humanitarian organizations, and private pharmacies, whereas the study from Yemen included government hospitals, PHCs, and private pharmacies, and the study from Afghanistan included public and private pharmacies. Second, the selection of medicines varied across all studies, with different emphasis on NCDs. Third, the formulations and dosages of medicines were not standardized across the three studies. Despite these limitations, the broadly similar patterns underscore the systemic impact of conflict on the availability of essential medicines.

The situation observed in Northern Syria in 2024 shares similarities with that of other fragile settings in the region, such as Iraq [38]. Both settings face challenges in ensuring that medicines reach those in need, due to a complex set of causes, including the collapse of the health and pharmaceutical systems, the impact of sanctions on both imports and local manufacturing, the physical insecurity that affects both supply of medicine and access to health facilities, and political and armed groups hijacking supply chains [38]. Furthermore, the high demand for medicines and the limited supply, combined with the inherent lack of market regulation in the midst of a conflict could have triggered the diversion of

essential medicines. This situation mirrors the practices seen in Iraq, where medicines were diverted and sold at inflated prices on the informal market for private profit, at the expense of public health. While the informal supply chain in Iraq reportedly involved even more complex networks of political parties, armed groups, and business people, the consequences of diversions and lack of regulatory oversight are equally deadly in both countries: the delayed (or absent) supply of medicine has major impact on the availability and prices of medicines in the formal health facilities, whether private or public. Moreover, substandard or falsified medicines may unnoticed enter the market, further compromising the health of vulnerable populations [38].

Our data also showed that epilepsy medicines had the lowest availability compared to those for cardiovascular disease and diabetes. These findings are consistent with other studies that have reported low availability of antiseizure medicines in fragile settings [34]. Epilepsy suffers from limited policy attention, both locally and internationally, despite the significant health and social consequences of untreated epilepsy. As observed in Northern Syria, stigmatization further compounds these challenges, limiting awareness and access to healthcare (including essential medicines), which is further exacerbated by the conflict. These observations underscore the urgent need for comprehensive strategies to improve access to epilepsy treatment and to address other neglected NCDs in conflict zones, including the integration of epilepsy care in humanitarian health interventions [34].

Our findings on access to insulin, whose global access is already severely compromised by the concentration of the market among multinational companies, show how poor access is exacerbated during conflicts. The low availability of Insulin reported in our study is in line with what has been reported about insulin's availability in low-resources settings in the region [39]. In conflict-affected areas, securing insulin is particularly challenging due to disruptions in healthcare services and transport. Traveling to healthcare facilities for insulin refills may be life-threatening in insecure regions, exacerbating the difficulty of disease management [40]. Additionally, medical centers in remote or conflict-affected areas frequently face stockouts of insulin, forcing patients to ration their supply or go without treatment entirely [40]. In the context of economic sanctions, pharmaceutical firms may be hesitant to export insulin to affected countries due to financial and legal uncertainties, despite medical products being formally exempt from sanctions [39]. This has been observed in Iran, where sanctions led to persistent shortages of insulin and diabetes management supplies. Due to the reluctance of insulin manufacturers to engage with Iran, the country was forced to pay insulin manufacturers double the price offered to other countries and more than five times the production cost for specific shipments of insulin pens, significantly raising prices and making insulin unaffordable for many patients [39]. Furthermore, insulin requires strict cold-chain storage, which is often disrupted in conflict settings: Power shortages, damaged infrastructure, and displacement create significant challenges in maintaining the required storage conditions [40]. Shortages of insulin impair diabetes management, increasing the likelihood of severe complications, including kidney failure, cardiovascular disease, vision loss, and diabetic foot amputation [40].

In the absence of a structured and regulated pharmaceutical system, the supply and pricing of medicines in Northern Syria are driven by the ability and opportunity of local suppliers to navigate economic, security, and logistical barriers. This situation results in significant variability in the availability and prices of medicines. Overall, the geopolitical situation hampers access to essential medicines, with the conflict disrupting supply chains and creating significant barriers to importing medicines from abroad. Geopolitical tensions also played a major role in influencing prices, as trade restrictions, sanctions, and regional instability drive up the cost of medicines. This situation is not unique to Syria, as similar trends have been observed in other conflict-affected countries, including Yemen and Afghanistan [34,35]. The impact of geopolitics on variability and fluctuations of medicine pricing highlights the need for different stakeholders acting on local, national, and international levels to adopt measures that minimize the impact of conflict on access to essential medicines. This should include reducing economic sanctions and conflict-related supply chain disruptions. To ensure access to medicines in northern Syria and other conflict-affected areas, stakeholders must uphold international humanitarian law, reinforcing compliance and accountability. Strengthening legal frameworks and fostering collaboration between health, legal, and political sectors are crucial to protecting civilians' right to essential medicines [20].

While this study examines medicine prices following the WHO/HAI methodology, affordability was not assessed as part of the WHO/HAI methodology [24]. This methodology estimates affordability based on the daily wage of the lowest-paid unskilled government worker (or an equivalent local benchmark) and calculates how many days' wages are required to purchase a full course of treatment. However, in northern Syria, the absence of a defined or regulated minimum wage limited the applicability of this method, making it challenging to generate meaningful affordability estimates. As a general reference point, the average monthly salary in Syria was reported to be around 81 USD in December 2020 [41], highlighting the severe economic hardship facing households over the past years. To complement this analysis, this project included qualitative interviews with NCD patients that explored their experiences with medicine affordability. Perceived affordability among patients will be analyzed and reported in a separate manuscript, providing further insights into the financial barriers to accessing medicines in this conflict-affected setting.

Although the study did not specifically focus on the quality of medicines, participants frequently expressed concerns about medicine quality during the interviews. The lack of regulation and oversight in the pharmaceutical sector has contributed to concerns about the possible prevalence of substandard and falsified medicines in Syria, as is common in conflict-affected regions. Studies from Afghanistan and Ethiopia reported that weak or absent regulatory mechanisms in fragile and conflict-affected settings allowed for the presence of poor-quality medicines in the local market [8,35]. Further research is needed to assess the actual quality of medicines in this context and to identify the facilitators necessary to ensure medicine quality in conflict-affected settings.

Overall, our study highlighted that the availability, prices and quality of medicines are interlinked factors that collectively determine access to medicines in a conflict area, and that are all negatively affected by the conflict. They are inextricably linked, with each factor affecting the others and influencing access to medicines overall [1].

Currently, the humanitarian situation in Syria is still dire, with a significant proportion of the population living in conflict-affected areas where access to essential medicines is severely limited [15]. Despite the emphasis on the right to access medicines under international humanitarian law frameworks, the reality on the ground is often different. Access to medicines has been threatened by various parties in the conflict, with medicines supplies being blocked or restricted for political reasons [20,38]. This situation is not unique to Syria, as similar challenges have been observed in other conflict zones, including Gaza, Sudan, and Yemen, where the delivery of essential medicines has been obstructed by conflicting parties [42–44]. The Syrian experience highlights the need for stronger adherence to the spirit of the international humanitarian laws and agreements, ensuring that those caught in conflict have uninterrupted access to essential healthcare including essential medicines. The international community must work together to ensure that humanitarian aid can reach those in need [15,20] while acknowledging the contribution that local health workers including pharmacists and medicine supplier play in long-conflicts, and exploring practical approaches for mutual collaboration.

The Assad's regime fall in December 2024 marked the end of more than five decades of human rights violations and destruction of the Syrian society [15]. While the transition presents a unique opportunity to address longstanding and more recent challenges, including the lack of access to essential medicines, the country remains in a highly fragile state. Warnings from experts about insecurity and retaliatory violence have materialized, with recent reports of mass civilian killings in Syria's northwest coast raising serious concerns about stability. Amnesty International has urged the new authorities to protect civilians and ensure accountability for these crimes. However, the failure to investigate and prevent such atrocities reinforces impunity and underscores that Syria remains unsafe [45]. The Syrian interim government now faces immense challenges in rebuilding the nation, particularly in restoring access to healthcare, which will require massive efforts and long-term commitment. With a fragile healthcare infrastructure and limited access to essential services, the health vulnerabilities of the Syrian people continue to deepen [15].

It is crucial to build on and strengthen local initiatives that aim to improve access to medicines, as these efforts have the potential to offer sustainable solutions in the long term. But short-term interventions are urgently needed to address the immediate gaps in access to healthcare and to essential medicines, particularly for NCDs such as epilepsy. While

supporting local pharmaceutical manufacturing remains a key-long term objective for improving access to medicines in Syria, it is unlikely to address the limited access to medicines in the short term. Therefore, immediate measures, such as improving supply chain mechanisms, and enhancing humanitarian aid coordination are necessary to mitigate the health crisis and prevent further deterioration. A comprehensive approach to NCDs should include the provision of quality-assured and affordable essential medicines, and the establishment of effective regulations. Lifting sanctions looks like an essential pre-requisite, as they keep on hindering the ability to import medicines and medical supplies into Syria, and had a major negative impact on the local pharmaceutical industry [15,46]. In the post-conflict era, the due attention must be given to NCDs prevention and treatment, and a stronger emphasis placed on ensuring that all people in Syria have adequate access to essential medicines.

Overall, we found limited literature to contrast and compare our findings. This highlights the need for more research, in fragile and conflict-affected settings, regarding the availability, prices and quality of essential medicines, as it is critical to provide evidence, advocate for the needs of communities, and inform policy makers. To be scientifically and societally relevant, such future research should adopt a collaborative approach, fairly involving local healthcare providers, policy-makers, communities, and concerned national and international organizations, to better understand the challenges and find sustainable solutions. Context-sensitive adaptation of research methods is essential, as the situation in conflict zones can change rapidly, requiring researchers to adapt their methodologies accordingly [47–49]. Therefore, revising standard methods to account for the unique challenges of conflict settings, such as the security concerns and the constant mobility of the population during armed conflicts [49], will improve the relevance and applicability of research findings. Collaborative efforts are critical for ensuring that the voices of affected communities are heard and that research outcomes can lead to actionable recommendations for improving access to essential medicines in conflict-affected areas.

## Strengths and limitations

The strengths of this study lie in its use of the WHO/HAI methodology, which was adapted to fit the conflict context. By combining both qualitative and quantitative methods, our mixed-methods approach provided a holistic perspective on the barriers to accessing essential medicines in the country. The involvement of a research advisory board further strengthened the study, both scientifically and ethically. However, there are also intrinsic limitations inherent to the WHO/HAI methodology. The cross-sectional nature of the survey, which relies on a single observation of surveyed pharmacies and healthcare facilities, limits the ability to assess the long-term availability of medicines or fluctuations in stock levels over time. Moreover, MPRs were not calculated, as the international reference price guidelines have not been updated since 2015, making them unsuitable for comparison; and affordability was not assessed, due to the absence of a defined minimum wage in northern Syria. This study did not include pediatric formulations, which limits the applicability of the findings to children with chronic conditions. Additionally, the involvement of the principal investigator (SA) in guiding respondents through virtual sessions may have introduced a potential for response bias, as respondents could have been unintentionally influenced by the investigator's (digital) presence or perceived expectations. While this approach aimed to ensure consistency and clarity in a challenging remote setting, it may have influenced how participants interpreted or responded to some items. The selection of surveyed essential medicines was necessarily restricted to cardiovascular, diabetes, and epilepsy medicines due to practical considerations related to feasibility of conducting the survey in conflict-affected areas. Consequently, other essential medicines for conditions outside these groups, although very important, could not be evaluated within the scope of this study.

Only the identified codes were translated into English to facilitate collaborative analysis among co-authors given the background and linguistic diversity in the research team. While this approach supported inclusive interpretation, it may have limited the wider team's access to the full contextual richness of participants' original narratives, introducing a potential risk of misinterpretation or analytic bias. While the use of snowball sampling may introduce some risk of sampling bias, the inclusion of a wide range of public and private health facilities across urban and rural areas in four regions of Northern Syria likely mitigates this limitation to some extent.

## Conclusions

This study highlighted severe challenges regarding the availability and pricing of essential medicines for NCDs in conflict-affected areas in Northern Syria. While the supply of medicine in this region relies on both humanitarian and private suppliers, NCD medicines are primarily sourced through local private suppliers, which are unable to meet the population's needs. Our data showed that the medicine supply is shaped by multiple conflict-related factors, including insecurity, movement restrictions, sanctions, and logistical disruptions. This situation resulted in low availability, significant price variation, and concerns about medicine quality. The particularly low availability of epilepsy medicines reflect the extreme neglect faced by patients with certain chronic conditions and underscores the urgent need for further research in access to care to neglected NCDs in conflict settings, using methodologies adapted to the inherent limitations of such environments, in order to understand and address the unique challenges faced by vulnerable populations during times of crisis.

Importantly, this study demonstrates that medicine supply systems in protracted conflict settings, while severely disrupted, do not disappear. Instead, they evolve, often into informal or opportunistic networks, but continue to function through alternative pathways. Recognizing this is the first step toward influencing these systems through targeted, realistic interventions. Cross-border dynamics also emerged as a key factor. Proximity to international borders, especially with Turkey and Iraq, created both risks of trafficking and opportunities for legal or semi-legal trade. These dynamics are already being leveraged by humanitarian actors who operate from neighboring countries. How supply systems develop in landlocked conflict zones remains a critical area for future research. In addition, participants reported the increasing use of mobile technologies, particularly WhatsApp, to share information on medicine availability, prices, and suppliers. These digital tools facilitate real-time coordination between pharmacists and wholesalers and offer promising opportunities for improving access in fragmented supply systems.

Formal control over medicine prices and flows is extremely limited, and attempts by local authorities to fix prices have largely failed. In the absence of effective regulation, operational collaboration, especially among pharmacists, wholesalers, and humanitarian actors. This may help to reduce shortages and improve reliability of supply. Looking ahead, priority must be given to developing coherent, context-sensitive policies to ensure access to essential medicines. While regulation remains important, it cannot substitute for policies that are grounded in the realities of local systems and resources. Pharmacists, who have played a key role throughout the crisis, should be supported and empowered as central actors in this effort.

The urgently-needed international support to rebuild Syria's healthcare system, will need to include the strengthening of local pharmaceutical and regulatory system, including targeted investments in research, monitoring and evaluation, with particular attention to the impact of sanctions and the need for safeguards that protect access to essential health services.

## Supporting information

**S1 Appendix. Interviews topic guides with study participants.**
(DOCX)

**S1 Checklist.**
(PDF)

## Acknowledgments

We extend our sincere gratitude to the members of our study advisory group for their invaluable guidance and contributions throughout this research. We thank Dr. Margaret Ewin for her advice on the selection of the essential medicines that we included in our study. We also thank Raphaela Delahaye and An Ielegems from our research office and support team for their dedicated assistance in this research project.

## Author contributions

**Conceptualization:** Saleh Aljadeeah, Karina Kielmann, Raffaella Ravinetto.

**Data curation:** Saleh Aljadeeah, Samah Fahed.

**Formal analysis:** Saleh Aljadeeah, Belen Tarrafeta, Samah Fahed, Raffaella Ravinetto.

**Funding acquisition:** Saleh Aljadeeah.

**Investigation:** Saleh Aljadeeah, Samah Fahed.

**Methodology:** Saleh Aljadeeah, Belen Tarrafeta, Karina Kielmann, Raffaella Ravinetto.

**Project administration:** Saleh Aljadeeah.

**Resources:** Saleh Aljadeeah.

**Software:** Saleh Aljadeeah.

**Supervision:** Karina Kielmann, Raffaella Ravinetto.

**Validation:** Saleh Aljadeeah, Belen Tarrafeta, Samah Fahed, Karina Kielmann.

**Visualization:** Saleh Aljadeeah, Samah Fahed.

**Writing – original draft:** Saleh Aljadeeah.

**Writing – review & editing:** Saleh Aljadeeah, Belen Tarrafeta, Samah Fahed, Karina Kielmann, Raffaella Ravinetto.

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
