## [Decision Letter · Decision Letter 0]

24 Jun 2025

PGPH-D-25-01253

Access to essential medicines for noncommunicable diseases during conflicts: the case cardiovascular diseases, diabetes and epilepsy in Northern Syria

Dear Dr. Aljadeeah,

Thank you for submitting your manuscript to PLOS Global Public Health. After careful consideration, we feel that it has merit but does not fully meet PLOS Global Public Health’s publication criteria as it currently stands. Therefore, we invite you to submit a revised version of the manuscript that addresses the points raised during the review process.

We look forward to receiving your revised manuscript.

Kind regards,

Hani Mowafi, M.D., M.P.H.

Academic Editor

Journal Requirements:

Additional Editor Comments (if provided):

Please address the comments of the reviewers and resubmit when ready.

Reviewers' comments:

Reviewer's Responses to Questions

**Comments to the Author**

1. Does this manuscript meet PLOS Global Public Health’s publication criteria?

Reviewer #1: Yes

Reviewer #2: Yes

2. Has the statistical analysis been performed appropriately and rigorously?

Reviewer #1: Yes

Reviewer #2: N/A

3. Have the authors made all data underlying the findings in their manuscript fully available (please refer to the Data Availability Statement at the start of the manuscript PDF file)?

Reviewer #1: No

Reviewer #2: No

4. Is the manuscript presented in an intelligible fashion and written in standard English?

Reviewer #1: Yes

Reviewer #2: Yes

Reviewer #1: Thank you for the opportunity to review your well conducted, well written study. I commend you for shedding the light on a global health topic of extreme importance and using a current, prolonged, ongoing conflict to provide valuable insights on reality of access to medicine in conflict zones.

The abstract clearly communicates the study's objectives, methods, key findings, and implications.

The background is rich, well-informed, and cites a substantial body of research. It effectively situates the study moving from a global perspective to a specific national context. The authors offer a concise description of the contemporary healthcare reality in Syria and how it has drastically changed because of the recent civil war. The Syrian context is presented as a case study that highlights challenges faced in conflict zones at large and thus offer valuable insights for concerned policy makers.

The authors chose to focus on three NCD: HTN, DM and Epilepsy. The choice is justified in the methods section.

The study adapts WHO/HAI methodologies with explained deviations to match the context of Northern Syria, accounting for specific limitations in healthcare infrastructure, regulatory frameworks, and licensing. The integration of quantitative and qualitative phases each with distinct protocols and rationale, strengthens the study's credibility and offers additional contextual depth to interpret the results.

The section addresses the ethical challenges of working in a conflict zone in a transparent and thorough way.

A potential for bias in data sampling should be acknowledged:

- Participants were guided by the investigators during the quantitative data collection (line 179). The supplement file lists the targeted questions. Clarify if one or more of the investigators were involved in these virtual sessions, and if the responders training was standardized. This step might introduce a potential for bias in the reported answers.

- The used grounded theory methodology adds credibility to the authors' data. It is noted that two investigators transcribed the interviews, but only translated the identified codes for the other research team members (line 223). This step might introduce a potential bias in data analysis.

- Snowball sampling method was probably necessary given the conditions on the ground in Northern Syria. However, a potential for sampling bias should be acknowledged.

- Consider adding a statement if any special requirements were issued by the Tropical Institute IRB given the fragile/conflict settings of the sampled population.

Results:

The qualitative results are clearly reported. The analysis covers several crucial and interrelated themes, which are organized in subsections in a logical manner. The themes are grounded in data with several supporting (translated) quotes from the participants.

This discussion section is rich and structured. It situates the study results into the broader topic of medicine access in a conflict-affected setting. Quantitative and qualitative results are effectively synthesized, with comparison to other conflict zones. The impact of systemic drivers such as geopolitical instability, and regulatory collapse on pharmaceutical access is aptly discussed. A call to action for further research into healthcare in conflict-zones highlights the added value of this study and offers suggestions for others to build on the author's experience.

Strength and Limitations are honestly discussed. I invite the authors to reflect on the additional limitations mentioned above in the methodology section.

A clear summary is provided in the conclusion section and emphasis on practical recommendations for policy makers and future research.

Other comments:

Line 79: health workers "fleeing" the country

Line 197: “international price reference guidelines has not been updated” → “have not been updated”

I commend the authors on their hard work.

Reviewer #2: Thank you for the opportunity to provide feedback to this important work.

Introduction: Lines 124-127. I imagine another primary goal of the research is to conduct a gap analysis for the Syrian health care system to allocate resources and better meet the treatment needs of the Syrian population.

Methods- Lines 163-165. Authors omitted epilepsy medication, which the clearly analyzed. Despite the explanation of the essential medicines, it’s unclear (and seems a little random) why the authors chose cardiovascular and anti-epilepsy drugs. The WHO 2023 list of essential medicine include: Non-steroidal anti-inflammatory medicine, psychotropic medicine, anti-allergic medicine, antidotes for poison, antibiotics/anti-viral/anti-fungal, anti-Parksinson, chemotherapy/immunodulators, contraceptives, vitamins/supplements, etc… It’s unclear why these were not examined.

Lines 225-227. It’s unclear who conducted the codes and analysis of the qualitative data. In line 225 it seems like the other co-authors conducted some of the coding. However, again, the PI’s involvement in the interviewing and coding does bias interpretation. How did multiple coders converge on themes? Were there consensus meetings? An adjudication process? Was any specialized software (NVivo or Atlas) used or were they collected in Excel? Was a semi-structured questionnaire developed? If so, then the questionnaire should be included in the supplemental tables.

In comparing northeast vs northwest and private vs public %, the authors could have conducted a simple pair wise chi squared analysis.

Results: Lines 304-314 and Table 2. No units are provided is it USD per 100 units, per tablet, per prescription. In the discussion, this needs to be contextualized to how the average Syrian can or can not afford. How much is the average salary and expenses of a Syrian? The cost by USD is not very meaningful with out context.

Discussion:

Lines 485-487 the authors only cite two studies from Yemen and Afghanistan without any detail. Please compare the data. Also, many more developing countries that are not in conflict have limited supplies of essential medications. It’s worth comparing Syria to those numbers.

Lines 643- I think that it’s a bit much to label this study as “comprehensive and nuanced”. Perhaps the mixed methods offered a “holistic” appreciation of the challenges Syrians face in accessing essential medicine.

Limitations include that the PI of the study conducted the interview. Most of the time interviews are not conducted by the PI to avoid bias on how the data is collected.

Minor feedback: Please check grammar throughout. For instance “low medicines”. Medicine can be itself a plural; throughout the text the authors use medicines, sometimes not appropriately. Language could be a bit tighter. For instance, “Our qualitive data suggested…” could be “Qualitative reports suggest…”

Figures/Tables: Line 84-96: consider adding a map to help the reader better understand the differential regional impact.

Table 1. It’s doesn’t make sense to take the decimal to the hundredth. Keep it to the tenth. It’s kind of a busy table. Consider color coding cells by low availability (30%–49%), and very low availability (<30%); maybe a pink and darker ping (red maybe too dark).

Table 2. Need to label that this is in USD.

**Do you want your identity to be public for this peer review?** For information about this choice, including consent withdrawal, please see our Privacy Policy

Reviewer #1: **Yes: ** Khaled Alok

Reviewer #2: No

---

## [Decision Letter · Decision Letter 1]

17 Oct 2025

PGPH-D-25-01253R1

Access to essential medicines for noncommunicable diseases during conflicts: the case cardiovascular diseases, diabetes and epilepsy in Northern Syria

Dear Dr. Aljadeeah,

Thank you for submitting your manuscript to PLOS Global Public Health. After careful consideration, we feel that it has merit but does not fully meet PLOS Global Public Health’s publication criteria as it currently stands. Therefore, we invite you to submit a revised version of the manuscript that addresses the points raised during the review process.

We look forward to receiving your revised manuscript.

Kind regards,

Helen Howard

Staff Editor

Journal Requirements:

Reviewers' comments:

Reviewer's Responses to Questions

**Comments to the Author**

Reviewer #2: (No Response)

Reviewer #3: (No Response)

publication criteria?

Reviewer #2: Partly

Reviewer #3: Yes

3. Has the statistical analysis been performed appropriately and rigorously?

Reviewer #2: N/A

Reviewer #3: Yes

4. Have the authors made all data underlying the findings in their manuscript fully available (please refer to the Data Availability Statement at the start of the manuscript PDF file)?

Reviewer #2: Yes

Reviewer #3: No

5. Is the manuscript presented in an intelligible fashion and written in standard English?

Reviewer #2: Yes

Reviewer #3: Yes

Reviewer #2: Methods-

Selection of essential medicine (Lines 167-178) still not adequately answered. The revision gave very vague answer "prior evidence of common prescription patterns, inclusion in the tenth Syraina National Essential medicine list, and input from advisory board". As I previously commented, the selection of cardiovascular, diabetic, and epilepsy meds seems random. The authors did not adequately address this. At this point, they should at least acknowledge in the limitation section that perhaps due to access, availability of information, time and resources, other essential medicine (such as those listed) could not be evaluated.

Discussion-

Regarding my feedback "Lines 485-487 the authors only cite two studies from Yemen and Afghanistan without any detail. Please compare the data. Also, many more developing countries that are not in

conflict have limited supplies of essential medications. It’s worth comparing Syria to those

Numbers" my point was that the discussion was restricted to comparing to Yemen and Afghanistan. However, most developing countries also have challenges to access to essential medications. Please do not restrict the discussion to these two countries. Expand the discussion by comparing to other countries' access to essential medications.

Table 1- need to explain what the different shades of purple mean

Figure 1- also need add a figure caption and o explain or label different colored regions

Reviewer #3: Introduction:

Line 111 – The term ‘insulin-dependent diabetic’ is obsolete and not used now. Please rephrase it to Type 1 Diabetes patients. The term ‘diabetic’ should also not be used.

The introduction is very long. Suggest to shorten the text by merging paragraphs 3, 4 and 5 with a focus on the problem statements. Lengthy explanations about the country are unnecessary.

Discussion:

555-559 – There is mention of a manufacturer’s name, “Novo Nordisk’. This name can be avoided as this is a scientific manuscript. Explaining the issue would suffice, or use a more general term eg, ‘the insulin supplier’.

609 – ‘At the moment of writing this paper’ is inappropriate as this is a scientific manuscript. Suggest rephrasing using other terms, eg, ‘Currently’ etc.

**Do you want your identity to be public for this peer review?** For information about this choice, including consent withdrawal, please see our Privacy Policy

Reviewer #2: No

Reviewer #3: **Yes: ** NAVIN KUMAR LOGANADAN

---

## [Decision Letter · Decision Letter 2]

11 Nov 2025

Access to essential medicines for noncommunicable diseases during conflicts: the case cardiovascular diseases, diabetes and epilepsy in Northern Syria

PGPH-D-25-01253R2

Dear Dr. Aljadeeah,

We are pleased to inform you that your manuscript 'Access to essential medicines for noncommunicable diseases during conflicts: the case cardiovascular diseases, diabetes and epilepsy in Northern Syria' has been provisionally accepted for publication in PLOS Global Public Health.

Best regards,

Julia Robinson

Executive Editor

Reviewer Comments (if any, and for reference):

Reviewer's Responses to Questions

**Comments to the Author**

Reviewer #2: All comments have been addressed

Reviewer #3: All comments have been addressed

publication criteria?

Reviewer #2: Yes

Reviewer #3: Yes

3. Has the statistical analysis been performed appropriately and rigorously?

Reviewer #2: N/A

Reviewer #3: Yes

4. Have the authors made all data underlying the findings in their manuscript fully available (please refer to the Data Availability Statement at the start of the manuscript PDF file)?

Reviewer #2: Yes

Reviewer #3: No

5. Is the manuscript presented in an intelligible fashion and written in standard English?

Reviewer #2: Yes

Reviewer #3: Yes

Reviewer #2: Authors adequately addressed my comments.

Reviewer #3: None. All the comments have been addressed.

**Do you want your identity to be public for this peer review?** For information about this choice, including consent withdrawal, please see our Privacy Policy

Reviewer #2: No

Reviewer #3: **Yes: ** NAVIN KUMAR LOGANADAN
